# LEARNING TO RECOGNIZE THE UNSEEN VISUAL PREDICATES

## ABSTRACT

Visual relationship recognition models are limited in the ability to generalize from finite seen predicates to unseen ones. We propose a new problem setting named predicate zero-shot learning (PZSL): learning to recognize the predicates without training data. It is unlike the previous zero-shot learning problem on visual relationship recognition which learns to recognize the unseen relationship triplets (`<subject, predicate, object>`) but requires all components (`subject`, `predicate`, and `object`) to be seen in the training set. For the PZSL problem, however, the models are expected to recognize the diverse even unseen predicates, which is meaningful for many downstream high-level tasks, like visual question answering, to handle complex scenes and open questions. The PZSL is a very challenging task since the predicates are very abstract and follow an extreme long-tail distribution. To address the PZSL problem, we present a model that performs compatibility learning leveraging the linguistic priors from the corpus and knowledge base. An unbalanced sampled-softmax is further developed to tackle the extreme long-tail distribution of predicates. Finally, the experiments are conducted to analyze the problem and verify the effectiveness of our methods. The dataset and source code will be released for further study.

## 1 INTRODUCTION

Visual relationship recognition (Johnson et al., 2015; Lu et al., 2016; Xu et al., 2017) aims to estimate the relationships between pairs of localized entities, i.e., performing the recognition of triplets `<subject, predicate, object>`. It structurally describes images, which provides rich semantic information of an image to many applications including visual question answering (VQA) (Li et al., 2018), image captioning (Yang et al., 2019) and image retrieval (Johnson et al., 2015). The relationship recognition methods are mainly supervised to recognize the entities and then combine various entities in pairs to identify predicates between them. There is an increasing interest in relationship zero-shot learning (ZSL) that learns to recognize the unseen relationship triplets, where the studies (Lu et al., 2016; Yu et al., 2017) on this ZSL problem setting assume the components (`subject`, `predicate`, and `object`) of the relationship triplet are seen. However, almost all of them only focus on dozens of frequent predicates and do not study the generalization of the seen predicates to the unseen ones.

In this work, we propose the predicate zero-shot learning (PZSL) problem setting focusing on recognizing the unseen predicates (no manual annotations or image samples). For example, given no instance of `chew` in the training data, the model is expected to recognize it during testing. Recognizing diverse even unseen predicates is significant for providing very rich relationship information, describing the complex scenes, and analogizing the known abstract concepts to the novel ones. The solution of the PZSL problem will greatly promote many downstream tasks, such as generating image caption with vivid predicates which are even unseen in the description corpus (image captioning) and answering the open questions (with novel predicates) on the complex scene (VQA).

Although zero-shot learning in image classification has received increasing attention (Larochelle et al., 2008), PZSL is not explored. Furthermore, the PZSL problem is more challenging in the following aspects. *a*) Recognizing predicates is difficult since predicates are often abstract not as specific as objects. Analogizing the seen abstract predicates to the unseen ones further escalates the difficulty. Many object ZSL methods (Lampert et al., 2014; 2009) adopt the pre-defined attributes of

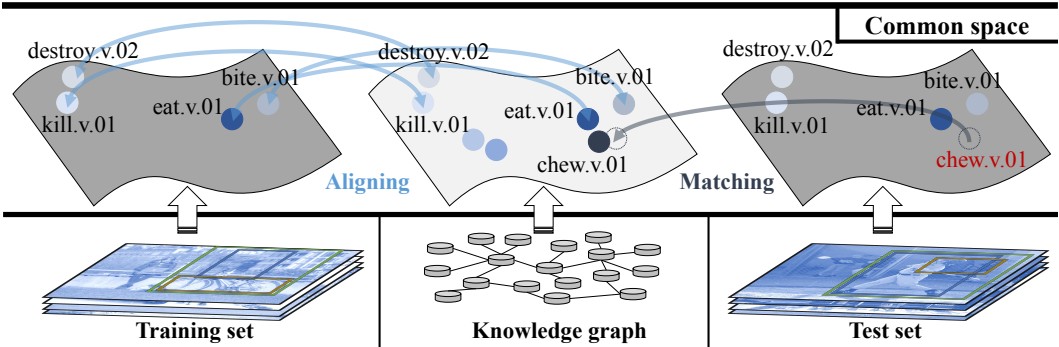

Figure 1: **A basic model for recognizing unseen visual predicates.** The visual data and knowledge graph's nodes are mapped into a common space by a visual and knowledge module respectively, where the sub-spaces from visual and knowledge module are named with visual feature and semantic embedding space correspondingly. Note that the visual predicate feature contains features of the subject, object and the union of them. The basic model contains two stages. First, the visual feature and semantic embedding space are *aligned* by taking the seen predicates in the training set as anchors, the so-called compatibility learning. Second, the samples in the test set are mapped into visual feature space and *matched* with the nearest predicate neighbor from the semantic embedding space.

objects to recognize the unseen object. However, it is difficult to define the attributes of predicates. *b*) Predicates of existing datasets follow an extreme long-tail distribution (92.26% predicates with the number of instances lower than 10 in Visual Genome (Krishna et al., 2017)). The statistics of the dataset (VG-Zero) in this work are shown in Fig. 4. Under this distribution, the model tends to collapse to output few frequent predicates. Note that if the infrequent predicates are not recognized, the unseen predicates are more unlikely to be recognized.

To address the PZSL problem, we introduce a basic model to perform compatibility learning (Frome et al., 2013; Akata et al., 2016; 2015) (Fig. 1), leveraging the linguistic priors from the corpus and knowledge base (Wang et al., 2018; Kampffmeyer et al., 2018). To represent the abstract predicates, we adopt the pre-trained word (sentence) vectors to initialize the predicates, connect them with linguistic relations defined in knowledge bases, and map them into a semantic embedding space (middle of Fig.1). A visual module is then applied to map paired image regions (left of Fig. 1) into a visual feature space. The visual feature and semantic embedding spaces fall in the common space (top of Fig. 1). During training, the visual feature and semantic embedding space are *aligned* with the seen predicates as anchors, i.e., a visual feature and semantic embedding labeled with the same predicate fall onto the same point/area in the common space. During testing, the samples in the test set are mapped into the visual feature space and *matched* with the nearest neighbor semantic embeddings of predicates (like chew). Furthermore, to tackle the long-tail distribution, an unbalanced sampled-softmax is developed to adjust the gradient penalty of the infrequent predicates.

The main contributions include: *a*) We define the predicate zero-shot learning (PZSL) problem and introduce the corresponding dataset (based on Visual Genome). *b*) We propose a basic model to address the PZSL problem by compatibility learning leveraging the linguistic priors from the corpus and knowledge base. *c*) We develop an unbalanced sampled-softmax for handling the extreme long-tail distribution of predicates. *d*) The dataset and source code of this work will be released.

## 2 RELATED WORK

**Visual relationships** have been studied from various aspects including statistical motifs (Zellers et al., 2018), entity-relationship dependencies (Xu et al., 2017), spatial priors (Dai et al., 2017), language statistics (Li et al., 2017). Almost all of them focus on recognizing dozens of the most frequent predicates. By contrast, our work explores to train a model with about 1000 predicates and test it with about 100 unseen predicates. The two most relevant problem settings are relationship zero-shot learning setting (Lu et al., 2016) and open vocabulary setting (Zhang et al., 2018). Lu et al. (2016) try to recognize the unseen relationships (e.g., <elephant, stand on, street>) by

transferring knowledge from similar relationships (e.g., <dog, stand on, street>). Note that all the test predicates and entities are seen in the training set. By contrast, the main difficulty in our problem setting is that all test predicates are unseen in the training set. Zhang et al. (2018) perform visual relationship recognition with an open vocabulary setting focusing on large-scale recognition problem without study on ZSL. The most relevant approach in visual relationship recognition is Graph R-CNN (Yang et al., 2018) which adopts a graph convolutional network (GCN) (Kipf & Welling, 2016) to capture contextual information between visual objects and relations. By contrast, we utilize GCNs to leverage the relations of categories to generate meaningful semantic embeddings.

**Zero-shot learning (ZSL).** To recognize unseen objects, compatibility learning frameworks (Frome et al., 2013; Fu et al., 2015; Fu & Sigal, 2016) map visual and semantic features into the common space and align the visual and semantic manifolds with the seen categories. During testing, these methods recognize the given visual feature by performing a nearest neighbor search on the semantic embeddings of the categories. Recent works (Wang et al., 2018; Kampffmeyer et al., 2018) utilize the linguistic relations between seen and unseen categories in a knowledge graph (KG) for ZSL. Wang et al. (2018) propose to train a GCN supervised by the classifier's weights of neural network. Our method can be considered as a hybrid of compatibility learning frameworks and knowledge graph based methods. The node embedding of KG is mapped into the same space with the visual feature.

**External knowledge bases (KB)**, such as Wikipedia and ConceptNet (Speer & Havasi, 2013), has been introduced in visual relationship recognition to provide linguistic and commonsense priors. Yu et al. (2017) extract the <subject, predicate, object> triplets from Wikipedia and leverage the statistics $\mathcal{P}(pred|sub, obj)$ to help recognizing the unseen relationship triplets. Baier et al. (2017) train a semantic model from a KB to improve the performance. Gu et al. (2019) takes the detected objects to retrieve on the ConceptNet to obtain a set of triplets to enhance the visual features. Unlike the prior works essentially using the statistics of <subject, predicate, object> from the external KB, our work leverages the *linguistic relations of predicates* defined in WordNet (Miller, 1992) to explicitly connect the predicates, such as <attack, is a hyponym of, fight>, for recognizing the unseen predicates.

## 3 PROBLEM SETUP

**Setting:** Let the full predicate vocabulary as $\mathcal{V}_{pred} = \mathcal{V}_{pred}^{tr} \cup \mathcal{V}_{pred}^{te}$ and entity vocabulary as $\mathcal{V}_{en}$, where $\mathcal{V}_{pred}^{tr}$ and $\mathcal{V}_{pred}^{te}$ are the training and test predicate vocabulary respectively, and "entity" refers to "subject" and "object". The training and test predicates are disjoint, i.e., $\mathcal{V}_{pred}^{tr} \cap \mathcal{V}_{pred}^{te} = \phi$. The dataset is denoted as $\mathcal{D} = \{(I_i, \langle b_{ij}^s, s_{ij}; p_{ij}; b_{ij}^o, o_{ij}\rangle)\}$, where $s_{ij}, o_{ij} \in \mathcal{V}_{en}$ denote subject and object labels of the $j$-th relationship in $i$-th image $I_i$ (the green box in Fig. 2 (A)), $b_{ij}^s$, $b_{ij}^o$ are the corresponding boxes of the subject and object, and $p_{ij} \in \mathcal{V}_{pred}^{tr} \cup \mathcal{V}_{pred}^{te}$ is the corresponding predicate label. Any image that contains a test predicate $p \in \mathcal{V}_{pred}^{te}$ is assigned to the test set $\mathcal{D}_{te}$, and only the regions with test predicate $p \in \mathcal{V}_{pred}^{te}$ are used for evaluation. The rest data is split into the training set $\mathcal{D}_{tr}$ and validation set $\mathcal{D}_{val}$. During testing, given an image and pair of subject and object boxes from the test set $\mathcal{D}_{te}$, the model (trained on $\mathcal{D}_{tr}$) recognizes a triplet <subject, predicate, object>, where the accuracy of predicates is in concern.

**Assumption:** We assume that, for any test predicate $p_{te} \in \mathcal{V}_{pred}^{te}$, there exists training predicate $p_{tr} \in \mathcal{V}_{pred}^{tr}$ having semantic association with $p_{te}$. For example, if chew is a test predicate, the predicates meaning an action using teeth (like bite) or intaking food (like eat) is expected to be included in the training set. Let the visual features and semantic embedding of bite are aligned, so does that of eat. As a result, the visual feature of chew, visual similar to that of bite and eat, is able to match to the predicates likes bite and eat in the semantic embedding space. To satisfy this assumption, the training predicates should be in large-scale to cover as much semantics as possible.

## 4 APPROACH

In this section, we first present the pipeline of our basic model, then the Simplified PinSage (a fast graph convolutional network) for propagating on the large-scale knowledge graph, and finally unbalanced sampled-softmax to handle the extreme long-tail distribution.

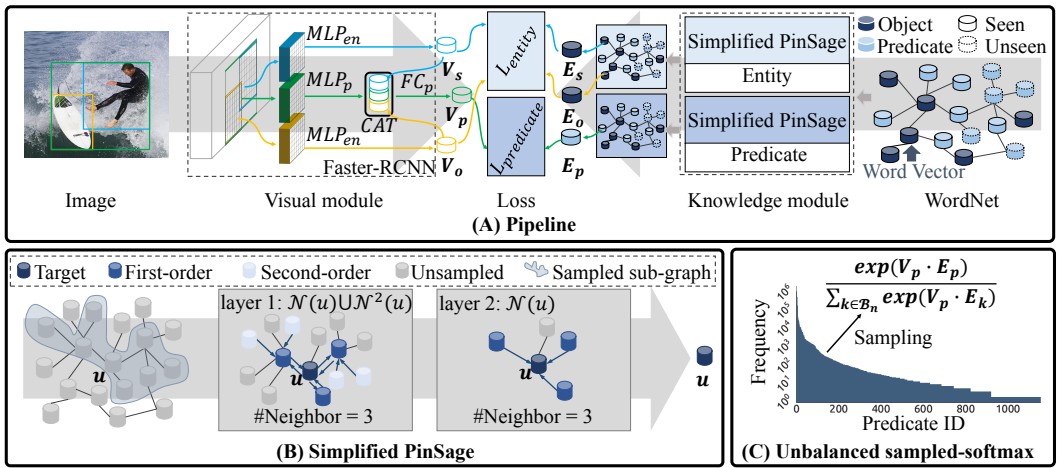

Figure 2: **A) The pipeline of our basic model.** The visual features (from the visual module) and corresponding semantic embeddings (from the knowledge module) are constrained to be close. **B) Simplified PinSage.** We sample the graph nodes in two aspects during training. Take 2-layer GCN as an example, for on-demand sampling, to get the final embedding of the target node $u$, the neighbors of $u$ are needed. Backtracking in this way, we only need first and second-order neighbors of $u$. For the neighbor-limit sampling, the number of each node's neighbors is limited by a constant. **C) Unbalanced sampled-softmax.** To tackle the long-tail distribution problem, a negative predicate batch $\mathcal{B}_n$ is sampled from the predicate distribution to calculate the loss function.

## 4.1 PIPELINE

The pipeline of our method consists of visual and knowledge modules, refer to Fig.2. They are modeled by a Faster R-CNN (Ren et al., 2017) and two GCNs (Kipf & Welling, 2016) respectively.

*Visual module* aims to extract visual features of entity and predicate. Given an image as input, the corresponding features of the subject, object and context region are cropped out, where context region refers to the union of subject and object regions. All these feature regions are ROI aligned (He et al., 2017) as ROI features with fixed size ($7 \times 7$). The ROI features of subjects (blue) and objects (yellow) are then mapped into visual entity features $V_s$ and $V_o$ by the same multilayer perceptron $MLP_{en}$. Furthermore, the ROI feature of context region (green box) is fed to $MLP_p$ and the output of which is fused with $V_s$ and $V_o$ to generate visual predicate feature $V_p$. All these features $V_s$, $V_o$ and $V_p$ will be aligned with the corresponding semantic embeddings $E_s$, $E_o$ and $E_p$.

*Knowledge module* aims to generate the meaningful semantic embeddings of the categories (including predicates and entities). To introduce the language prior and implicit association of the categories, the embeddings of the categories are initialized with word (sentence) vectors pre-trained on a large-scale corpus (like GloVe (Pennington et al., 2014)). The categories are then connected by the linguistic relations (defined by a KB, WordNet) to build a knowledge graph. Note that many categories are not directly connected, but can be indirectly connected through categories within the KB outside the dataset. Thus the knowledge graph contains a huge number of nodes and provides rich linguistic information. Taking the knowledge graph as input, the semantic embeddings of predicate $E_p$ and entity $E_{en}$ are generated by graph convolutional networks $GCN_p$, $GCN_{en}$ respectively. For this part, the Simplified PinSage is introduced for fast processing on the large-scale graph, refer to § 4.2.

*Loss function* is defined as a summation of the entity and predicate terms as follows:
$$\mathcal{L} = \mathcal{L}_{entity} + \mathcal{L}_{predicate}, \tag{1}$$
where $\mathcal{L}_{predicate}$ is designed with the proposed unbalanced sampled-softmax for tackling the long-tail distribution of predicates, refer to § 4.3, while $\mathcal{L}_{entity}$ is a negative log likelihood with softmax:
$$\mathcal{L}_{entity} = \mathbb{E}_{(V_c,c)}[-log\frac{exp(V_c \cdot E_c)}{\sum_{k \in \mathcal{V}_{en}} exp(V_c \cdot E_k)}], \tag{2}$$
where $c \in \mathcal{V}_{en}$ is the label of the visual entity feature $V_c$.

## 4.2 Simplified PinSage

Simplified PinSage is adopted to map the knowledge graph into an embedding space for PZSL, see Fig.2 (B). Inspired by PinSage (Ying et al., 2018), the propagation algorithm is divided into three steps: message passing, skip shortcut and normalization. We denote $FC_{W,B} \circ x = Wx + B$, and the process of graph propagation is formulated as follows:

$$z_u^{k-1} = \frac{1}{|\mathcal{N}(u)|} \sum_{v \in \mathcal{N}(u)} ReLU(FC_{W_{t_k}, B_{t_k}} \circ h_v^{k-1}), \tag{3}$$

$$\tilde{h}_u^k = ReLU(FC_{W_{c_k}, B_{c_k}} \circ [z_u^{k-1}, h_u^{k-1}]), \tag{4}$$

$$h_u^k = \tilde{h}_u^k / \|\tilde{h}_u^k\|_2, \tag{5}$$

where Eq. (3), (4) and (5) indicate message passing, skip shortcut and normalization respectively, $\mathcal{N}(u)$ in Eq. (3) denotes the neighbor set of $u$ ($u$ also falls into $\mathcal{N}(u)$), $[\cdot, \cdot]$ in Eq. (4) means "concatenate", $h_u^0$ is the initial embedding, and $h_u^k$ is output of $k$-th graph propagation layer. For $n$-layer GCN, $F_{W_{out}, B_{out}}$ takes $h_u^n$ as input to get the final embedding $E$ lying in the same space with the visual feature $V$. Note that the main computation in forward propagation is related to the number of edges (Eq. (3)). It is too computation and space consuming to perform propagation on the whole knowledge graph with about 2.2 billion edges. The graph sampling technology in web-scale recommender system (Ying et al., 2018; Eksombatchai et al., 2018) is introduced as an solution.

**On-demand sampling**. For generating embeddings of mini-batch categories, we only need to sample a necessary sub-graph as input to GCN ($GCN_p$ or $GCN_{en}$), avoiding propagating on the whole knowledge graph. Take 2-layer GCN as an example. Only first and second-order neighbors are needed to compute the final embedding of the target nodes. Refer to Fig. 2 (B), to get the embedding of $u$ (deep blue), the second layer of GCN needs the embeddings (outputs of the first layer) of neighbors (blue) of $u$ for message passing, i.e., $\mathcal{N}(u)$. To get the embeddings of $\mathcal{N}(u)$ in the first layer, the neighbors of the node set $\mathcal{N}(u)$ are needed (light blue), i.e., $\mathcal{N}^2(u)$. Hence, we only sample a sub-graph containing the node set $\mathcal{N}(u) \cup \mathcal{N}^2(u)$. In general, to get final embeddings of nodes batch $\mathcal{U}$ for $n$-layer GCN, only 1st to $n$-th order neighbors are needed, i.e., $\cup_{i=1}^n \mathcal{N}^i(\mathcal{U})$.

**Neighbor-limit sampling**. Many nodes of the knowledge graph contain a large number of neighbors, which makes computation and space consumption uncontrollable. To further reduce the consumption, we limit the number of neighbors per node to a threshold $\tau$, i.e., randomly sampling $\tau$ neighbors. Refer to Fig. 2 (B), the unsampled neighbors (gray) do not contribute to the propagation. For testing, all neighbors are sampled to calculate the final embeddings, and we only need to propagate once to obtain the final semantic embeddings. The experiments in § 5 show that neighbor-limit sampling can be considered as a dropout-like operation for greatly avoiding overfitting.

Thus, to obtain embedding of $k$ predicates from $n$-layer GCN, the number of edges are not greater than a relaxed upper bound $k\tau^n$, where $k, \tau \leq 100$ and $n \leq 3$. Thus we have $k\tau^n \leq 10^8 \ll 2.2 \times 10^{10}$.

## 4.3 Unbalanced sampled-softmax

A variant softmax function is proposed to measure the similarity between visual features and semantic embeddings, inspired by sampled softmax (Jean et al., 2014) in machine translation. For training of predicate recognition, if all elements of training vocabulary are sampled as negative embeddings, such as standard softmax, the visual features may always match the few most frequent predicate embeddings since the long-tail distribution of the predicates in the dataset. The impact of this distribution on zero-shot learning is devastating. To tackle this problem, we propose an unbalanced sampled-softmax (USS):

$$S_i = \frac{exp(V_p \cdot E_i)}{\sum_{k \in \mathcal{B}_n} exp(V_p \cdot E_k)}, \mathcal{B}_n \sim \mathcal{P}_{pred}, \tag{6}$$

where $V_p$ is the visual feature whose predicate category is $p$, $E_i$ is the corresponding semantic embedding of predicate $i$. Unlike sampled softmax that adopts the pre-divided sub-vocabulary as negative categories, the negative predicates $\mathcal{B}_n \subset \mathcal{V}_{tr}$ is sampled from the predicate distribution

$\mathcal{P}_{pred}$. It is possible that $\mathcal{B}_n$ includes the GT predicate $p$. Finally, the loss function of predicate recognition is in the form of

$$\mathcal{L}_{predicate} = \mathbb{E}_{(V_p,p)}[-log(S_p)]. \tag{7}$$

The sampling method is vital for Eq. (7). The uniform sampling, degrading into an estimated version of softmax, does not help with long-tail distribution. We design a sampling method to ensure the recognition of predicates with fewer samples so that the model can be further generalized to recognize unseen predicate categories. The idea is that the fewer categories appear as positive categories, the less they are sampled as negative categories. We adopt the frequency of predicates as the probability $\mathcal{P}_{pred}$ to sample the negative predicates, see Fig. 2 (C). This sampling method handles the long-tail distribution problem by adjusting the gradient of the infrequent predicates.

Let $h_i = V \cdot E_i$, the gradient of $S_p$ w.r.t $h_i$ is discussed as follows:

$$\frac{\partial S_p}{\partial h_i} = \begin{cases} S_p(\mathbf{1}(p = i) - S_i) & \text{if } i \in \mathcal{B}_n \\ \mathbf{1}(p = i)S_i & \text{if } i \notin \mathcal{B}_n. \end{cases} \tag{8}$$

Frequent predicates often fall into the first case in Eq. (8), which is the same as standard softmax. To the opposite, infrequent predicates always fall into the second case that the reward is increased when it is GT ($p = i$) and that there is no punishment when it is a negative predicate ($p \neq i$).

## 5 EXPERIMENTS

In this section, we start by discussing the datasets, knowledge graph, and implementation details. We then perform the ablation studies to verify the components of our model and visualize our results.

**VG-zero dataset.** We introduce a new dataset based on the latest released Visual Genome dataset (VG v1.4) (Krishna et al., 2017) which contains 108,077 images with 21 relationships (triplets) on average per image. We manually cleaned up the box annotations in the same way with Xu et al. (2017). Since the original annotation is noisy, 1155 synsets in WordNet are used to replace the original predicate categories as regularization, where the correspondence between the original categories and synsets is provided in the VG dataset. About $10\%$ of predicates (105 predicates) are selected as test vocabulary. The frequency of the selected predicates falls in a range from 10 to 300, where the lower bound 10 is set to guarantee the quality of test set for the infrequent labels are noisy, and the upper bound 300 follows the rule that categories in test set should be least populated or rare (Xian et al., 2017) in zero-shot learning. Images annotated with predicates in test vocabulary are selected as the test set (containing 4350 images). We then randomly select 5000 images as the validation set with the rest as the training set. Similar to predicates, the entity categories are also replaced by 7k+ synsets. In addition, we use hypernym relationships to cluster entity categories into 96 categories since the entity synsets are still so specific that include numerous names and object recognition is not our focus. For example, categories like woman, father are clustered into the person category.

**Knowledge graph.** The knowledge graph $\mathcal{G}(\mathcal{V}, \mathcal{E})$ is built from WordNet (Miller, 1992), where $\mathcal{V}$ and $\mathcal{E}$ are nodes set and edges set respectively. Synsets (synonym set) in WordNet are nodes in $\mathcal{G}$. Edge $\langle u, v \rangle$ is added into $\mathcal{E}$ if $u$ and $v$ have one of the following relationships: hypernym, hyponym, part meronym, part holonym, substance meronym, substance holonym, entailment, substance holonym and sharing lemmas. Notice that the self-loop will be included by the "sharing lemmas" relationship. Finally, the resulted knowledge graph contains 101,260 nodes and about 2.2 billion edges.

**Implementation details.** For all experiments, the model is trained for 150k iterations with batch size set to 4. We set the learning rate as $2e^{-3}$ and is reduced by 0.1 times at the 100k and 130k, respectively. We adopt the warmup strategy (Goyal et al., 2017) at the beginning. ResNet50 (He et al., 2016) is used as a backbone network with weights pre-trained on COCO (Lin et al., 2014), which is fixed during training. Images are resized such that their short edge is 800 pixels. For the knowledge graph, we use the definitions of synsets as the input of off-the-shelf language models to generate the initial embeddings. More specifically, we use the word (GloVe) and sentence (InferSent (Conneau et al., 2017)) embedding methods to get the initial embeddings. For the word embedding method, we take every word of the definition as a token to GloVe and average all the word embeddings to get the 300-D embedding. For the sentence embedding method, the whole definition is used as the input of InferSent to get the 4096-D embedding. The common space and hidden features in GCN are 512-D.

## 5.1 Ablation study

Table 1: Accuracy of unseen predicate recognition.

| NO. | Propagation | Initial embedding | Loss | Hits@k (%) | | | | | | | |
| | | | | Generalized setting | | | Traditional setting | | | | |
| | | | | 5 | 10 | 20 | 1 | 2 | 5 | 10 | 20 |
|---|---|---|---|---|---|---|---|---|---|---|---|
| 1 | W/O KG | GloVe | $\|\mathcal{B}_n\| = 10$ | 0.0 | 0.0 | 0.0 | 2.5 | 5.2 | 13.0 | 23.1 | 37.9 |
| 2 | 1-layer GCN | GloVe | $\|\mathcal{B}_n\| = 10$ | 1.9 | 4.7 | 10.1 | 5.8 | 10.7 | 20.2 | 32.3 | 48.5 |
| 3 | 2-layer GCN | GloVe | $\|\mathcal{B}_n\| = 10$ | **4.3** | 7.0 | 11.3 | 7.5 | 12.2 | 22.2 | 33.2 | 48.7 |
| 4 | 3-layer GCN | GloVe | $\|\mathcal{B}_n\| = 10$ | 1.9 | 3.9 | 8.4 | 5.3 | 9.3 | 19.9 | 33.3 | 49.1 |
| 5 | 2-layer GCN | GloVe | Softmax | 0.0 | 0.0 | 0.0 | 2.2 | 4.2 | 10.8 | 19.0 | 33.0 |
| 6 | 2-layer GCN | GloVe | $\|\mathcal{B}_n\| = 5$ | 3.2 | 6.5 | 10.6 | 6.5 | 10.9 | 20.1 | 32.5 | 48.1 |
| 7 | 2-layer GCN | GloVe | $\|\mathcal{B}_n\| = 20$ | 4.1 | **7.4** | **11.8** | **8.9** | **13.0** | 21.3 | 32.0 | 49.5 |
| 8 | 2-layer GCN | GloVe | $\|\mathcal{B}_n\| = 50$ | 2.0 | 4.7 | 9.9 | 5.5 | 10.4 | 21.7 | 32.9 | 48.6 |
| 9 | 2-layer GCN | GloVe | $\|\mathcal{B}_n\| = 100$ | 1.3 | 3.8 | 8.5 | 4.6 | 8.8 | 18.3 | 29.5 | 46.2 |
| 10 | $\tau = 5$ | GloVe | $\|\mathcal{B}_n\| = 10$ | 1.9 | 4.7 | 10.1 | 4.2 | 8.9 | 20.2 | 31.7 | 48.6 |
| 11 | $\tau = 20$ | GloVe | $\|\mathcal{B}_n\| = 10$ | 1.2 | 4.1 | 8.7 | 3.9 | 8.3 | 19.8 | 32.6 | 50.5 |
| 12 | $\tau = 50$ | GloVe | $\|\mathcal{B}_n\| = 10$ | 2.5 | 5.3 | 9.2 | 4.5 | 9.1 | 18.1 | 30.8 | 48.8 |
| 13 | $\tau = 100$ | GloVe | $\|\mathcal{B}_n\| = 10$ | 1.9 | 4.0 | 7.8 | 4.4 | 8.1 | 17.5 | 31.4 | 50.3 |
| 14 | 2-layer GCN | Normal | $\|\mathcal{B}_n\| = 10$ | 0.0 | 0.0 | 0.0 | 1.4 | 2.4 | 5.0 | 10.2 | 19.6 |
| 15 | 2-layer GCN | InferSent | $\|\mathcal{B}_n\| = 10$ | 4.1 | 7.0 | 11.4 | 7.3 | 12.4 | **23.1** | **35.8** | **53.5** |
| 16 | | Random guess | | 0.1 | 0.8 | 1.7 | 0.9 | 1.9 | 4.7 | 9.5 | 19.0 |

Table 2: Accuracy of recognition of triplets with unseen predicates.

| Methods | Hits@k (%) | | | | | | | |
| | Generalized setting | | | Traditional setting | | | | |
| | 5 | 10 | 20 | 1 | 2 | 5 | 10 | 20 |
|---|---|---|---|---|---|---|---|---|
| W/O KG | 0.0 | 0.0 | 0.0 | 1.1 | 2.3 | 7.8 | 12.5 | 20.1 |
| Softmax | 0.0 | 0.0 | 0.0 | 0.9 | 1.9 | 5.0 | 9.9 | 17.5 |
| Ours | **1.3** | **2.8** | **5.8** | **3.2** | **6.0** | **11.6** | **18.0** | **26.1** |
| Random guess | $4.7e^{-7}$ | $9.4e^{-7}$ | $1.9e^{-6}$ | $1.0e^{-6}$ | $2.1e^{-6}$ | $5.2e^{-6}$ | $1.0e^{-5}$ | $2.1e^{-5}$ |

We evaluate methods with Hit@k on generalized and traditional setting (Tab. 1 and 2 ): Given any test region pair, we check whether the GT label falls within predicted categories with top k score. If yes, the sample is counted as a $Hit$. The Hit@k is $\frac{\#Hit}{\#Test\ region\ pair}$. The methods are only required to $Hit$ the unseen predicates ignoring the subjects and objects in Tab 1, while they are required to $Hit$ the full relationship triplets in Tab 2. For the *generalized setting*, the methods search on the seen and unseen predicates vocabulary $\mathcal{V}_{tr} \cup \mathcal{V}_{te}$ for inferrence. For the *traditional setting*, the methods only search on the unseen predicates vocabulary $\mathcal{V}_{te}$. Note that the generalized setting is more challenging since the search space of the generalized setting is significantly larger than that of the traditional one. Observing Tab. 1 and 2, we can draw the following conclusions.

**Knowledge graph** prior is critical to our algorithm. As shown in the first row of the Tab. 1, a simple 2-layer MLP for semantic embedding, which neglects the relationship of predicates, cannot identify the unseen predicates in the generalized setting (0 accuracies) and can only obtain a lower recognition rate on the traditional setting. An 1-layer GCN that simply considers node relationships can already deliver significant performance gains (row 2 of Tab. 1), with a large margin (10.6 (%)) on Hit@20 traditional setting. The results show that by modeling the explicit connection between predicates, the knowledge can be effectively transferred from seen categories to unseen ones, which results in a performance boost on the unseen predicate. The same conclusion can be obtained from Tab. 2.

**The unbalanced sampled-softmax** effectively tackles the long-tail distribution problem. Refer to row 5 in Tab. 1, adopting the softmax loss function results in the worst performance since the outputs collapse into few frequent predicates and could not be generalized to novel predicates. By contrast, adopting the frequency of predicates as sampling probability has obvious advantages with a 15.7 (%) increase on Hit@20 traditional setting (comparing row 3 and 5 in Tab. 1). The same conclusion can be made by comparing row 2 and 3 in Tab. 2.

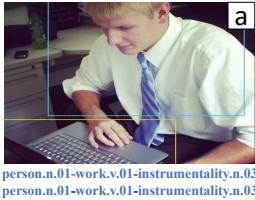 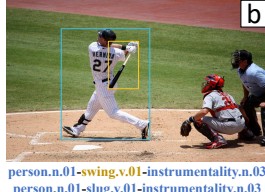 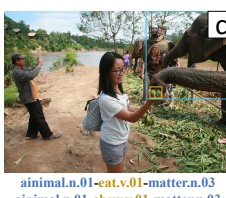 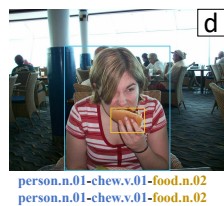

Figure 3: The images with paired located entities are on the top, where the blue and yellow boxes represent subject and object respectively. The results of the generalized and traditional setting are at the first two rows (correct: blue, wrong: yellow), while the ground truth triplets are at the last row.

**The number of negative categories** should be moderate (row 3 and 6∼9). Note that the larger $|\mathcal{B}_n|$, the higher probability that the infrequent predicates fall into $\mathcal{B}_n$, e.g., if $|\mathcal{B}_n| = |\mathcal{V}_{tr} \cup \mathcal{V}_{te}|$, the USS degenerates to softmax. When $|\mathcal{B}_n| < 100$, the performance of models with different $|\mathcal{B}_n|$ become similar, while setting with $|\mathcal{B}_n| = 20$ approach the best accuracy (49.5 (%) on Hit@20 traditional setting). However, the performance of the version with $|\mathcal{B}_n| = 100$ drops significantly (46.2 (%)).

**Embedding initialization** is necessary, but the impact of different embedding methods on performance is minor. Refer to row 14 in Tab.1, we adopt noise following normal distribution to initialize node embeddings, whose results are almost the same as random guessing (row 16). The initial embedding method clusters the semantically similar categories, which implicitly connects seen and unseen categories. These implicit connections are helpful for zero-shot learning. While using different embedding methods results in little difference: the InferSent method enjoys advantages over the GloVe method on the traditional setting with Hit@5∼20 but with no advantage in other evaluations.

**The numbers of neighbors and layers** result in negligible performance. For traditional setting, observing row 10∼13 in Tab. 1, the impact of $\tau$ is limited. Similarly, GCNs with different layers achieve almost equal accuracies, refer to row 2∼4. For the generalized setting, row 4 (3-layer GCN) and 13 (with 100 neighbors) achieve relatively low accuracy, reducing 2.9 (%) on Hit@20 generalized setting. It can be interpreted that a large number of training parameters and complete neighborhood information in the knowledge module make the model overfit the training vocabulary.

### 5.2 QUALITATIVE RESULTS

Fig.3 shows the results of our method, where output is in the form of `<subject, predicate, object>`. The recognition of predicates includes generalized and traditional settings shown at row 1 and 2 (blue/yellow font represent correct/wrong), while the ground truth triplets are displayed in the last row. The case $a$ is completely correct in both settings. In case $b$, our method makes a mistake in the generalized setting while the result of the recognition (`swing.v.01`) is close to the ground truth (`slug.v.01`). This case shows that predicate zero-shot learning in the generalized setting is hard for semantically similar categories across training and test vocabularies. The case $c$ is so confusing that even humans can make misjudgment. In case $d$, our method determines the predicate as `chew.v.01` and outputs a more appropriate answer than the ground truth on the recognition of the object. In conclusion, the predicate zero-shot learning is challenging, but our method is effective.

## 6 CONCLUSIONS & FUTURE WORK

In this work, we define a predicate zero-shot learning problem and propose a solution by mapping visual features and semantic embeddings (in knowledge graph) into the common space. Furthermore, graph sampling strategies are introduced for accelerating graph propagation, and an unbalanced sampled-softmax is proposed for tackling the long-tail distribution. Finally, we plan to explore the following future work for this problem. $a$) Consistency of differences of visual feature and semantic embedding space can be considered to model the cross-modal analogy. $b$) A semantic-aware negative sampling of predicate categories is a solution worth exploring. $c$) Learning the visual predicate as a translation vector (Bordes et al., 2013; Wang et al., 2014; Lin et al., 2015) from subject to object such as VtransE (Zhang et al., 2017) deserve attempted.

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

# A    ADDITIONAL DETAILS OF VG-ZERO DATASET

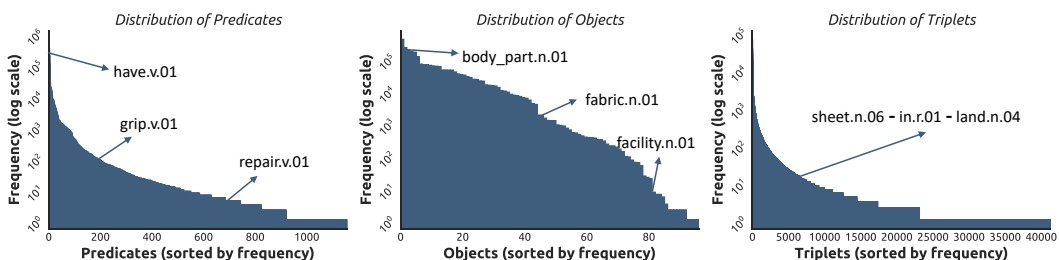

Figure 4: Distribution of predicates, objects and relationship triplets.

**The process of relabeling.** The relationship instances in Visual Genome are not only labeled with the "predicate" (in natural language, e.g., `has`), but also with the synset (node of WordNet, e.g., `have.v.01`) corresponding to the "predicate". We filter out all relationship instances without labeling synset and use the synset to replace the original predicate, e.g., "have.v.01" replace "has". The category of the entity is treated in the same way.

**The statistics** of predicates, objects, and triplets are shown in Fig. 4. Predicates follow an extreme long-tail distribution. The instances' number of the top 5 predicates count for 80% of the total instances (1155 predicates).

# B    VISUALIZATION OF THE EMBEDDINGS

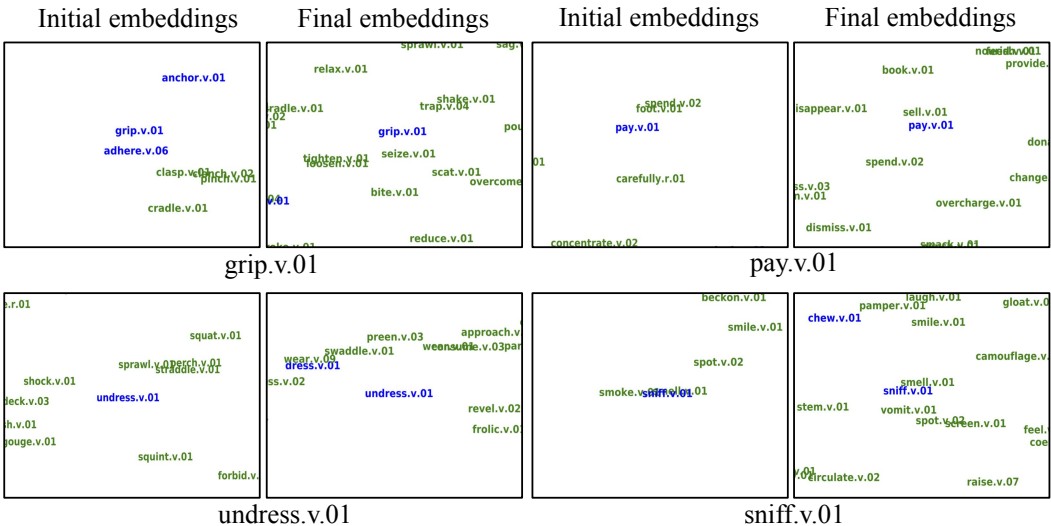

Figure 5: Comparing the initial embeddings and the final embeddings. The unseen predicates are marked in blue, while the seen predicates are marked in green.

We use t-SNE to visualize the initial and final embeddings of predicates, where the initial embeddings are from GloVe and the final embeddings are generated by our knowledge module. Refer to Fig. 5, the blue font means the unseen predicates while the green font means the seen ones. We focus on the neighborhood of unseen predicates. It can be concluded that our knowledge module leverages explicit connections to "pull together" the semantically similar predicates.

