# OpenReview forum: "Learning to Recognize the Unseen Visual Predicates"
_ICLR.cc/2020/Conference — Reject_

### Official Review · AnonReviewer1 · 2019-10-23
**Official Blind Review #1**

**Rating:** 6

**Review:**

This paper creates a new task for zero-shot learning of predicates (specifically in cases where the individual predicate components have never been seen in the training, rather than the more traditional setting where the full s-v-o relationship is unseen but each component is).  They create a new subset of the Visual Genome dataset specifically targeted towards predicting unseen predicates.  They also provide results using a knowledge graph (in this case WordNet), to integrate linguistic and visual features for prediction.  Interestingly, their pipeline introduces a new softmax variant, the unbalanced sampled softmax, which addresses the problem of over-predicting common predicates.

I generally tend towards accepting this paper.  The reasons being that this paper has a few strong contributions: (1) they design a new task set-up with data they selected and cleaned from VG, (2) new modelling pipeline with empirical analysis backing the modeling choices, and (3) new softmax variant.

A few comments about things that could be strengthened or addressed further:
- There could be more meaningful comparison to other zero-shot learning algorithms.  Even if they are not fully comparable because they were originally meant for a slightly different zsl set-up, it would be nice to have more baselines from external work.
- Why was the unbalanced sampled softmax was being used for only predicate prediction and not entity prediction?
- It wasn’t totally clear to me whether all of the verbs/entities were in WordNet and/or Glove.  If not, can the authors clarify what the overlap was and how this might be affecting performance?
- As noted by the authors, there are some cases in Figure 3 where humans would consider the answer to be confusing or might actually prefer the machine response.  Can human performance on this task could be measured?  Perhaps humans could possibly evaluate a subset of the machine vs. gold answers?

Minor Edits:
- Related work: WordNet is misspelled in the last line
- Figure 3: please consider using colors other than red/green, this is not readable for color-blind readers
- Section 5.2: “The case c is confusing that even” →  “The case c is so confusing that even”
- Section 5.2: “and output a more appropriate” --> “and outputs a more appropriate”
- In the references: the Devise paper by [Frome et al 2013] is listed twice
- In the references: the first author of ConceptNet5 should be Robyn Speer

**Experience Assessment:**

I have read many papers in this area.

**Review Assessment: Checking Correctness Of Derivations And Theory:**

N/A

**Review Assessment: Checking Correctness Of Experiments:**

I assessed the sensibility of the experiments.

**Review Assessment: Thoroughness In Paper Reading:**

I read the paper at least twice and used my best judgement in assessing the paper.

---

> ### Author Response · Authors · 2019-11-15
> **Response to Reviewer #1**
>
> We thank the reviewer for tending towards accepting our work and mentioning that our work has “a few strong contributions”. The “minor edits” are done in the revised version.
>
> *** Response to the things that could be strengthened or addressed further ***
>
> Q 1. There could be more meaningful comparison to other zero-shot learning algorithms.
> A 1. Thanks for the great suggestion. Due to the time limit, we will add such results after the rebuttal period.
>
>
> Q 2. Why was the unbalanced sampled-softmax was being used for only predicate prediction and not entity prediction?
> A 2. (Paragraph 2, Sec 5, P6). The recognition of the entity is a full-supervised task. Since entity recognition is not the focus of this work, we have controlled the number of object categories by clustering with hypernym relationships. After that, the distribution of entities is relatively balanced. In this setting, the softmax has a good recognition effect.
>
>
> Q 3. It wasn’t totally clear to the reviewer whether all of the verbs/entities were in WordNet and/or Glove.
> A 3.
> a. All of the verbs/entities are in WordNet.
> In this problem setting, we let all the predicate categories to exist in WordNet.
> An original annotation in Visual Genome is shown in the follows:
> ++++++++++++++++++++++++++++++++++++++++++++++++++++++++++
> "predicate": "has",
> …
> "relationship_canon":
> [ { "synset_name": "have.v.01", "synset_definition": "have or possess, either in a concrete or an abstract sense" } ]
> ++++++++++++++++++++++++++++++++++++++++++++++++++++++++++
> The relationship instances are not only labeled with the “predicate” (in natural language), but also with the synset (node of WordNet) corresponding to the “predicate”. We filter out all relationship instances without labeling synset and used the synset to replace the original predicate, e.g., “have.v.01” replace “has”. Thus, all the predicate labels are in WordNet.
>
> b. Are all of the verbs/entities in Glove?
> Take the above sample as an example, the definition of “have.v.01” is “have or possess, either in a concrete or an abstract sense”. We take all the words of the definition sentence as tokens to obtain the word embeddings and average all the word embeddings. We ignore the word not in GloVe.
>
> Q 4. Can human performance on this task be measured?
> A 4. Thanks for the great suggestion. The experiment is ongoing.
> We are calling on undergraduates to perform the experiment and the process of the user study is proposed as:
> 1. A test image with two boxes is shown to the user.
> 2. The user types the keywords into the interface.
> 3. The system searches the predicates with the input keywords.
> 4. The user selects several output predicates as the answers.
> In the final version paper, we will report the human experimental results and details.

---

### Official Review · AnonReviewer2 · 2019-10-23
**Official Blind Review #2**

**Rating:** 3

**Review:**

The paper considers the task of predicting visual predicates (e.g., eat, bite, take) between pairs of entities. In particular, the paper focuses on the zero-shot setting where the test predicates are unseen during training. The model uses linguistic prior from a knowledge graph (WordNet):
with graph embedding (fast GCN), unseen predicates are embedded based on the information propagated from seen predicates. The model is trained so that the visual feature vector and the correct predicate embedding are nearby in the joint embedding space. The method was evaluated on a zero-shot split of the Visual Genome dataset.

Overall, as a task and dataset paper, the paper should have sold the task more by highlighting its special properties. While the task of predicting unseen predicates is interesting, the setting and the technique are similar to previous work on predicting other types of unseen labels (e.g., unseen objects, as referenced in the paper). The new task could still be interesting if it presents different challenges (e.g., maybe predicates are more ambiguous than objects, or visual predicates are harder to embed). But from the description in the paper, most of the challenges seem to also exist in other zero-shot settings (long-tail distribution; large space of labels). The paper could benefit from providing examples or statistics that demonstrate the challenges of the task.

The proposed method looks correct but is a rather direct application of existing methods. The experiment setup looks OK. Based on the error analysis, the labels look very noisy and subjective, but this seems to be a common problem in the visual predicate prediction task (hence the recall-based evaluation).

Additional questions:

- The provided examples in the error analysis look pretty tricky; e.g., "swing" and "slug" are judged as different. How well would a human do on this task?

- How much would Hit@k be if the test label is seen during training (not zero-shot)?


**Experience Assessment:**

I have read many papers in this area.

**Review Assessment: Checking Correctness Of Derivations And Theory:**

I assessed the sensibility of the derivations and theory.

**Review Assessment: Checking Correctness Of Experiments:**

I assessed the sensibility of the experiments.

**Review Assessment: Thoroughness In Paper Reading:**

I read the paper thoroughly.

---

> ### Author Response · Authors · 2019-11-15
> **Response to Reviewer #2**
>
> We thank the reviewer for the appreciation of this paper and have followed the valuable advice to revise the paper.
>
> *** Response to the main questions ***
> Q 1. The special properties and challenges of predicate zero-shot learning.
> A 1.
> Properties of PZSL are summarized as follows:
> a) PZSL is very necessary since the difficulty of labeling. To avoid exhaustively traversing all pairs ($O(N^2)$) of entities ($O(N)$), the labels of predicates (VG) are just extracted based on image descriptions. In this process, the predicates are seriously missing. It reflects the necessity of PZSL.
>
> b) PZSL is significant for downstream tasks. VRD is often used to generate a scene graph (SG, edge: predicate, node: entity) for visual reasoning (like VQA). As long as the SG is restricted to a closed vocabulary, the reasoning engine [1,2] is inevitably difficult to cope with the open questions with very diverse unseen predicates or entities. For visual reasoning, PZSL may make the state transition (the edge of SG) more generalized. (Paragraph 2, Sec 1)
>
> Challenges of PZSL are summarized as follows:
> c) Recognizing predicates is very difficult. 1). It is difficult to represent predicate. On the one hand, predicates are often abstract not as specific as objects. On the other hand, the representation of predicates also depends on that of subjects and objects. How to effectively represent the abstract predicates is still a hard problem. 2) Analogizing the seen abstract predicates to the unseen ones further brings in new difficulty. Furthermore, unlike many object ZSL methods, they adopt the pre-defined attributes of objects to recognize the unseen object. However, it is difficult to define the attributes of predicates. (Paragraph 3, Sec 1)
>
> d) The long-tail distribution of predicates is significantly more severe than that of objects. As suggested by the reviewer, the extreme long-tail distribution of predicates is presented in Fig. 4 of Appendix A. In the VG-Zero dataset, the instances' number of the top $5$ predicates count for about $80\%$ of the total predicate instances. Under this distribution, the model tends to collapse to output few frequent predicates. As an initial solution to PZSL, we have to give priority to solving the extreme long-tail distribution (unbalanced sampled-softmax). (Paragraph 3, Sec 1)
>
> [1] Hudson D A, Manning C D. Learning by abstraction: The neural state machine.
> [2] Mao J, Gan C, Kohli P, et al. The Neuro-Symbolic Concept Learner: Interpreting Scenes, Words, and Sentences From Natural Supervision.
>
>
> Q 2. The proposed method looks correct but is a rather direct application of existing methods.
> A 2.
> a. We are the first to propose the unbalanced sampled-softmax to handle the extreme long-tail distribution. The most similar method Sampled Softmax, proposed in NLP for the large-scale vocabulary, fails under the extreme long-tail distribution.
>
> b. The graph sampling is borrowed from the recommender system domain after sufficient problem modeling and is very suitable to deal with large-scale KG for VRD problem.
>
> *** Response to the additional questions ***
> Q 3. How well would a human do on this task?
> A 3. Thanks for the great suggestion; the experiment is ongoing.
> We are calling on undergraduates to perform the experiment, where the process of the user study is proposed as:
> 1. A test image with two boxes is shown to the user.
> 2. The user types the keywords into the interface.
> 3. The system searches the predicates with the input keywords.
> 4. The user selects several output predicates as the answers.
> In the final version paper, we will report the human experimental results and details.
>
> Q 4. How much would Hits@k be if the test label is seen during training (not zero-shot)?
> A 4. We test the model in Row 7, Table 2. The results of the generalized setting are as follows:
>
> Test set                     | Hit@5 Hit@10 Hit@20
> ====================================
> Unseen predicates |   4.1         7.4       11.8
> Seen predicates      |  67.5        75.7     81.6
> Mixed predicates    |  62.0        69.7     75.4
>
> “Seen predicates” means we only test on the region pairs that labeled with the seen predicates.
> “Mixed predicates” means we test on all the region pairs from the test images.

---

### Official Review · AnonReviewer3 · 2019-10-24
**Official Blind Review #3**

**Rating:** 6

**Review:**

Title: Good work, requires some edits.

1. Summarize:

This paper proposes a new problem setting in visual relation detection which is called “Predicate Zero-shot Learning (PZSL)”. They provide a clear motivation and description of this setting. They propose a solution to this problem which leverages linguistic priors and knowledge bases. Furthermore they propose an unbalanced sampled-softmax to tackle the long tail distribution of predicates.

2. Clearly state your decision. One or two key reasons for this choice.

I will go for a weak accept for the paper at this stage. (+) I think the proposed problem setting is well-motivated and useful. Also, (+) the proposed initial solution to this problem is interesting. However, (-) they propose a “fast graph convolution network” which seems to be precisely equivalent to a PinSage.  Also, (-) the paper requires to be polished as it lacks clarity.

3. Main discussion

My first argument is: I’m not sure why the authors have changed the name of PinSage and just mentioned that “their” “Fast Graph Convolution Network” is “inspired” from PinSage. To me it looks exactly the same. If there are any differences, it should be stated clearly. In fact, I would not be against using PinSage as a part of their approach. However, trying to rename it without clear reasons is not a good idea.

My second argument is that the paper lacks clarity in writing (for detailed suggestions please refer to comments and feedbacks). Specially the evaluation section lacks details and clarity: a) In the beginning of this section (page 7), the authors talk about “generalized” and “traditional” settings without properly defining them. b) The descriptions for Table 1 and Table 2 fail to provide enough details to help understand the difference between the results in these two tables (one of them states “Accuracy of unseen predicate recognition” and the other one “Accuracy of recognition of triplets with unseen predicates”).

4. Comments and feedback.

Introduction:

Paragraph one in the:
1. The relationship recognition methods are mainly supervised “that” → “to”.
2. last line: …. and do not study “on generalizing” → “the generalization of”.

Paragraph two:
1. no manual annotations or “real samples” → “image samples”. (a real sample is ill-defined)
2. For example, no instance of chew → For example “given” no instance of chew.

Paragraph three:
1. … is difficult since predicates are often abstract not as specific →   is difficult since predicates are often abstract “and” not as specific.
2. Furthermore, unlike many object ZSL methods … → This line to the end is very complicated and hard to understand.

Related Works:

1. Visual Relationships: I would cite “Graph R-CNN for scene graph generation” since it is the most relevant work regarding the similarity of pipeline (using GCNs).
2. External Knowledge bases  (KB): I would cite “Improving Visual Relationship Detection using
Semantic Modeling of Scene Descriptions” since it is one of the most relevant works using knowledge graph modelings to improve visual relation detection.

Problem Setup:

Do you plan to provide the proposed dataset splits so others can work on this setting? I consider this very important given your paper’s contribution. Maybe it is better if it is also mentioned in the paper.

Pipeline:

1. Paragraph 2: … the output of which is fused with …: Given Figure 2, it does not seem like $V_p$ is being created by fusing $V_s$ and $V_o$. It looks more like it is extracted directly from the image (union of bounding boxes).

2. In Figure 2: In the representation of Pipeline (A), the graph is colored by dark blue for objects and light blue for predicates. The represented graphs show Object to Object and Predicate to Predicate connections which I’m not sure if it is correct. Shouldn’t we always have a light blue between every pair of dark blue connections?

Evaluation:

1. Please consider the mentioned points in the Main Discussions.
2. In Table 1, I suggest re-naming “embedding” to “initial embedding”.
3. In Table 1, Hit@k should be Hits@k.
4. Please define the metrics clearly (Hits@k).

Extra: I have a question regarding the ablation studies with GloVe, Normal and InferSent initialization. The question is whether this initialization is necessary? It seems like in the setting “W/O KG”, even though the embeddings are initialized with GloVe, there is no gain (all of the Hits@k values are 0.0). So GloVe embedding without KG bring no external semantic knowledge? Then why use them? Regarding that, I can see that a GCN, initialized with normally distributed embeddings (row 14 in Table 1) has given 0.0 accuracies, but I find this very counter intuitive, as graph convolution layers already have trainable weights capable of compensating for the lack of ‘proper’ initialized embedding and getting 0.0 does not make sense to me.

Conclusions and Future Work:

1. two lines before the last: please use “\citep”.



**Experience Assessment:**

I have published one or two papers in this area.

**Review Assessment: Checking Correctness Of Derivations And Theory:**

I carefully checked the derivations and theory.

**Review Assessment: Checking Correctness Of Experiments:**

I assessed the sensibility of the experiments.

**Review Assessment: Thoroughness In Paper Reading:**

I read the paper thoroughly.

---

> ### Author Response · Authors · 2019-11-15
> **Response to Reviewer #3 (Response to comments and feedback, 2/2)**
>
> Q 5. Do you plan to provide the proposed dataset splits so others can work on this setting?
> A 5. Yes, we will release the dataset as well as the splits setting. Source code will also be released to facilitate further researches.
>
>
> Q 6 (Extra). “I have a question regarding the ablation studies with GloVe, Normal and InferSent initialization. The question is whether this initialization is necessary?”
> A 6. The experimental results in Row 1 (only with GloVe)，14 (only with WordNet) and 3 (with both) in Table 2 are analyzed as follows.
>
> The generalized setting of PZSL is a very difficult test setting. Note that we only constrain the visual features matching the embeddings of the seen predicates during training. Thus, the visual features are more likely to match the embeddings of seen predicates. As the GT labels in the test set only containing the unseen predicates, all the seen predicates are wrong answers during testing, which makes the task difficult (both versions “only with GloVe” and “only with WordNet” get 0.0 accuracy).
>
> Observing on the traditional setting (only search on the unseen predicates for inference), the version “only with GloVe” (37.9 Hit@20) significantly outperforms the version “only with WordNet” (19.6 Hit@20). Furthermore, combining these two versions (“with both”) will achieve significant improvement (48.7 Hit@20).
>
> The reason why the version “only with WordNet” get a poor performance can be analyzed as follows:
> The GCN is just trained to align the visual features and semantic embeddings of the seen predicates. In this process, we have NO explicit constraints to build a good embedding space, such as putting together the semantically similar predicates. Due to the randomness of the initial embedding, the initial embeddings of two semantically similar predicates may be quite different. In this setting, the GCN is not explicitly supervised to pull closer the two predicates in the embedding space. Therefore, there is no guarantee that the randomly initialized embedding is able to achieve good results with WordNet.
>
> Q 7. “The represented graphs show Object to Object and Predicate to Predicate connections which I’m not sure if it is correct?”
> A 7. Object to-Object and Predicate-to-Predicate connections do exist in WordNet.
> Actually, the synsets (nodes in WordNet) are used to replace the original predicate categories. As shown in the following example, we use “have.v.01” to replace “has” as the predicate category.
>
> An original annotation in Visual Genome is shown in the follows (Please focus on the “synset_name”):
> ++++++++++++++++++++++++++++++++++++++++++++++++++++++++++
> "predicate": "has",
> …
> "relationship_canon":
> [ { "synset_name": "have.v.01", "synset_definition": "have or possess, either in a concrete or an abstract sense" } ]
> ++++++++++++++++++++++++++++++++++++++++++++++++++++++++++
>
> WordNet contains the linguistic relationships of the synsets. For example, “have.v.01” is the hypernym of “stock.v.01” (the definition of “stock.v.01” is “have on hand”). Thus, there is an edge between “have.v.01” and “stock.v.01”. This explains why Predicate-to-Predicate connections exist. Similarly, Object-to-Object connections also exist.
>
> As mentioned in Paragraph “Knowledge graph”, Sec 5, we use the following linguistic relationships (provided in WordNet) to build edges:
> hypernym,
> hyponym,
> part meronym,
> part holonym,
> substance meronym,
> substance holonym,
> entailment,
> substance holonym
> and sharing lemmas.
>
> As long as there is one linguistic relationship between two nodes, there will be one edge.

---

> ### Author Response · Authors · 2019-11-15
> **Response to Reviewer #3 (Main discussion, 1/2)**
>
> We thank the reviewer for the valuable comments and constructive suggestions on our paper. The minor errors have been revised and the related references have also been added.
>
> *** Response to the main discussion ***
> Q 1. Comparing “fast GCN” and “PinSage”. Why the authors have changed the name of PinSage and just mentioned that “their” “Fast Graph Convolution Network” is “inspired” from PinSage?
> A 1. Thanks for the valuable comments. “Fast GCN” is a simplified version of “PinSage”. The name “Fast GCN” tends to cause confusion so that we have changed the name to “Simplified PinSage”.
>
> The components in PinSage that we don't use include:
> 1. Producer-consumer minibatch construction
> 2. Efficient MapReduce inference
> 3. Importance pooling
> 4. Curriculum training
>
> The components in PinSage that we use include:
> 1. The architecture of GCN.
> 2. The node sampling strategy in minibatch, which we call “on-demand sampling”.
> 3. The neighbor sampling strategy is a little different:
> PinSage: adopt the “importance pooling” to sample neighborhoods.
> Ours: We randomly sample neighborhoods.
>
>
> Q 2. “generalized” and “traditional” settings lack clarity.
> A 2. The revised definition is shown as follows:
> The seen predicates are the predicates that appear in the training set, while the unseen predicates are those not in the training set. For the generalized setting, the methods search on the seen and unseen predicates vocabulary $\mathcal{V}_{tr} \cup \mathcal{V}_{te}$ for inference. For the traditional setting, the methods only search on the unseen predicates vocabulary $\mathcal{V}_{te}$. Note that the generalized setting is more challenging since the search space of the generalized setting is significantly larger than that of the traditional one.
>
>
> Q 3. Please define the metrics clearly (Hits@k).
> A 3. The definition of the metrics is revised as:
> We evaluate methods with Hit@k on generalized and traditional settings (Table 1 and 2): Given any test region pair, we check whether the GT label falls within the predicted categories with top k score. If yes, the sample is counted as a $Hit$. The Hit@k is $\frac{\#Hit}{\#Test\ region\ pair}$.
>
>
> Q 4. The descriptions for Table 1 and Table 2 fail to provide enough details to help understand the difference between the results in these two tables.
> A 4.
> In Table 1 (Accuracy of unseen predicate recognition), the methods are only required to “hit” the unseen predicates ignoring the subjects and objects.
>
> In Table 2 (Accuracy of recognition of triplets with unseen predicates), the methods are required to “hit” the full relationship triplets (subject, predicate, object). Only when all the elements of the triplet are “hit” simultaneously can it be counted as a “hit”.

---

### Author Response · Authors · 2019-11-15
**To all reviewers**

We thank reviewers for the very constructive comments and appreciation. We have revised the paper based on most of the comments. Due to the time limit, some experiments are ongoing and will be added in the final version. Please refer to the updated paper and answers to each reviewer.

---

### Decision · Program_Chairs · 2019-12-19

**Decision:**

Reject

**Comment:**

The paper proposes a new problem setting of predicate zero-shot learning for visual relation recognition for the setting when some of the predicates are missing, and a model that is able to address it.

All reviewers agreed that the problem setting is interesting and important, but had reservations about the proposed model. In particular, the reviewers were concerned that it is too simple of a step from existing methods. One reviewer also pointed towards potential comparisons with other zero-shot methods.

Following that discussion, I recommend rejection at this time but highly encourage the authors to take the feedback into account and resubmit to another venue.